# Heparan sulfate-dependent phase separation of CCL5 and its chemotactic activity

**Xiaolin Yu[1†], Guangfei Duan[1†], Pengfei Pei[1], Long Chen[1], Renji Gu[1], Wenrui Hu[1], Hongli Zhang[1], Yan-Dong Wang[1], Lili Gong[2], Lihong Liu[2], Ting-Ting Chu[3*], Jin-Ping Li[4,5*], Shi-Zhong Luo[1*]**

[1]State Key Laboratory of Chemical Resource Engineering, College of Life Science and Technology, Beijing University of Chemical Technology, Beijing, China; [2]Institute of Medical Science, China-Japan Friendship Hospital, Beijing, China; [3]Greater Bay Biomedical InnoCenter, Shenzhen Bay Laboratory, Shenzhen, China; [4]Beijing Advanced Innovation Centre for Soft Matter Science and Engineering, Beijing University of Chemical Technology, Beijing, China; [5]Department of Medical Biochemistry and Microbiology, University of Uppsala, Uppsala, Sweden

**\*For correspondence:**
tingtingchu@szbl.ac.cn (T-TC);
jin-ping.li@imbim.uu.se (J-PingL);
luosz@mail.buct.edu.cn (S-ZhongL)

[†]These authors contributed equally to this work

**Abstract** Secreted chemokines form concentration gradients in target tissues to control migratory directions and patterns of immune cells in response to inflammatory stimulation; however, how the gradients are formed is much debated. Heparan sulfate (HS) binds to chemokines and modulates their activities. In this study, we investigated the roles of HS in the gradient formation and chemoattractant activity of CCL5 that is known to bind to HS. CCL5 and heparin underwent liquid–liquid phase separation and formed gradient, which was confirmed using CCL5 immobilized on heparin-beads. The biological implication of HS in CCL5 gradient formation was established in CHO-K1 (wild-type) and CHO-677 (lacking HS) cells by Transwell assay. The effect of HS on CCL5 chemoattractant activity was further proved by Transwell assay of human peripheral blood cells. Finally, peritoneal injection of the chemokines into mice showed reduced recruitment of inflammatory cells either by mutant CCL5 (lacking heparin-binding sequence) or by addition of heparin to wild-type CCL5. Our experimental data propose that co-phase separation of CCL5 with HS establishes a specific chemokine concentration gradient to trigger directional cell migration. The results warrant further investigation on other heparin-binding chemokines and allows for a more elaborate insight into disease process and new treatment strategies.

## eLife assessment

How the triplicate interaction between chemokines with both GAGs and G protein-coupled receptors (GPCR) works and how gradients are created and potentially maintained in vivo are poorly understood. The authors provide **solid** evidence to show phase separation can drive chemotactic gradient formation. The paper is a **useful** advance in the field of chemokine biology.

## Introduction

Gradients of signaling molecules are ubiquitous in embryonic development and cellular activities, modulating cell migration, proliferation, and survival (*Griffith et al., 2014*; *Yamamoto et al., 2022*; *Yu et al., 2009*). Chemokines, belonging to a ~45-member family of small (8–12 kDa) proteins, are signaling molecules that can induce distinct cellular chemotaxis in response to inflammatory stimuli

in a concentration-dependent gradient (*Weber et al., 2013*). Chemokines play critical roles in regulating cell migration in a wide range of biological activities, for example, developmental, homeostatic, and inflammatory/pathological processes (*Dyer et al., 2016*). It is established that chemokines are secreted from source cells and immobilized on glycosaminoglycans (GAGs) including heparan sulfate (HS) in the extracellular matrix (*Proudfoot et al., 2003*). The interaction of GAG–chemokine forms an immobilized gradient to provide direction to cell movement (*Proudfoot et al., 2017*). Understanding how these chemokine gradients are formed and maintained is fundamental to identifying how they direct cell migration and proliferation. Straightforward molecular diffusion is often proposed as a possible mechanism (*Kicheva et al., 2007*; *Schier and Needleman, 2009*; *Yu et al., 2009*). However, it is assumed that more complicated mechanisms are involved, which remains to be elucidated.

CCL5 (also known as RANTES, for 'regulated on activation normal T cell expressed and secreted') (*Roscic-Mrkic et al., 2003*) is an inflammatory chemokine that recruits a wide variety of leukocytes, including monocytes, granulocytes, and T cells, as well as mast cells and dendritic cells, through chemokine gradients (*Roy et al., 2015*; *Weber et al., 2013*). CCL5 is composed 68 amino acids that reversibly self-assembles into high-MW oligomers, up to >600 kDa. This highly basic protein binds heparin with high affinity (*Mulloy, 2005*) and the oligomerization of CCL5 for the formation of gradient is modulated by GAGs on cell surface (*Proudfoot et al., 2003*).

Liquid–liquid phase separation (LLPS) driven by weak interactions between multivalent biomolecules was shown to be an important mechanism by which mesoscale structures of the condensates can form within the cell (*Xue et al., 2019*) and on the cell surface (*Xue et al., 2022*). Within phase-separated condensates, biomolecules are often mobile and interchanged between the dense and light phases. Based on reported information and our previous study, we have studied the implications of HS on CCL5 gradient formation and its chemotactic activity. Both in vitro and in vivo experiments demonstrate that co-phase separation of CCL5 with HS establishes specific chemokine concentration gradients for chemotactic activity.

## Results

### Co-phase separation of CCL5 and heparin

To illustrate the phase separation property of CCL5 in liquid, recombinant protein expressed in *E. coli* (*Figure 1—figure supplement 1A*; *Proudfoot et al., 1996*) was labeled with organic dye Cy3 (CCL5-Cy3) and mixed with heparin at different ratios. After 30-min formation of droplets was examined by fluorescence under confocal microscopy. At the ratio of 20:1 CCL5-Cy3 to heparin, few tiny droplets were detected, indicating phase separation. With increased ratio of CCL5-Cy3:heparin to 1:20 (20 μM CCL5-Cy3:1 μM heparin), large and round droplets were formed (*Figure 1A, B*). Further increases in heparin concentration (ratio 1:50) prevented droplet formation as a 'reentrant' behavior. Since heparin is a linear repetitive structure with rich negative charges, the LLPS seems following a similar mechanism as RNA in the phase separation of many RNA-binding proteins (*Ghosh et al., 2019*). Longer incubation revealed that the droplets were dynamic and some became fused to form larger droplets (*Figure 1C*) that displayed a quick recovery speed in the fluorescence recovery after photobleaching (FRAP) (*Figure 1D*). The heparin-induced phase separation of CCL5 was attenuated at high salt concentrations as illustrated by incubation at different KCl concentrations in KMEI buffer (0.5M KCl, 0.1M imidazole , 10mM EDTA, 20mM MgCl2, KMEI; pH 7.1) (*Figure 1—figure supplement 2A, B*). In comparison, a mutant (A22K-CCL5) with higher positive charge density showed stronger interaction with heparin (*Brandner et al., 2009*) and formed aggregates instead of phase separation with heparin in KMEI buffer (*Figure 1—figure supplement 2C, D*). These results indicated that heparin-induced LLPS of CCL5 was based on weak electrostatic interactions between the negatively charged polysaccharide and the basic protein of CCL5, while strong ionic interaction has abolished this effect.

An earlier study showed complex formation of CCL5 with a heparin disaccharide, which proposed the BBXB motif of CCL5 playing essential role in the interaction between the chemokine and the disaccharide (*Shaw et al., 2004*). In light with this finding, we mutated [44]RKNR[47], a conserved positively charged BBXB motif of CCL5, to hydrophobic alanine motif [44]AANA[47] (*Proudfoot et al., 2001*), which is expected to reduce the interaction with heparin. As anticipated, the phase separation of the mutant [44]AANA[47]-CCL5 in solution containing heparin formed much smaller and fewer droplets in comparison to WT CCL5 (*Figure 1E, F*). The electrostatic interaction feature was further demonstrated by

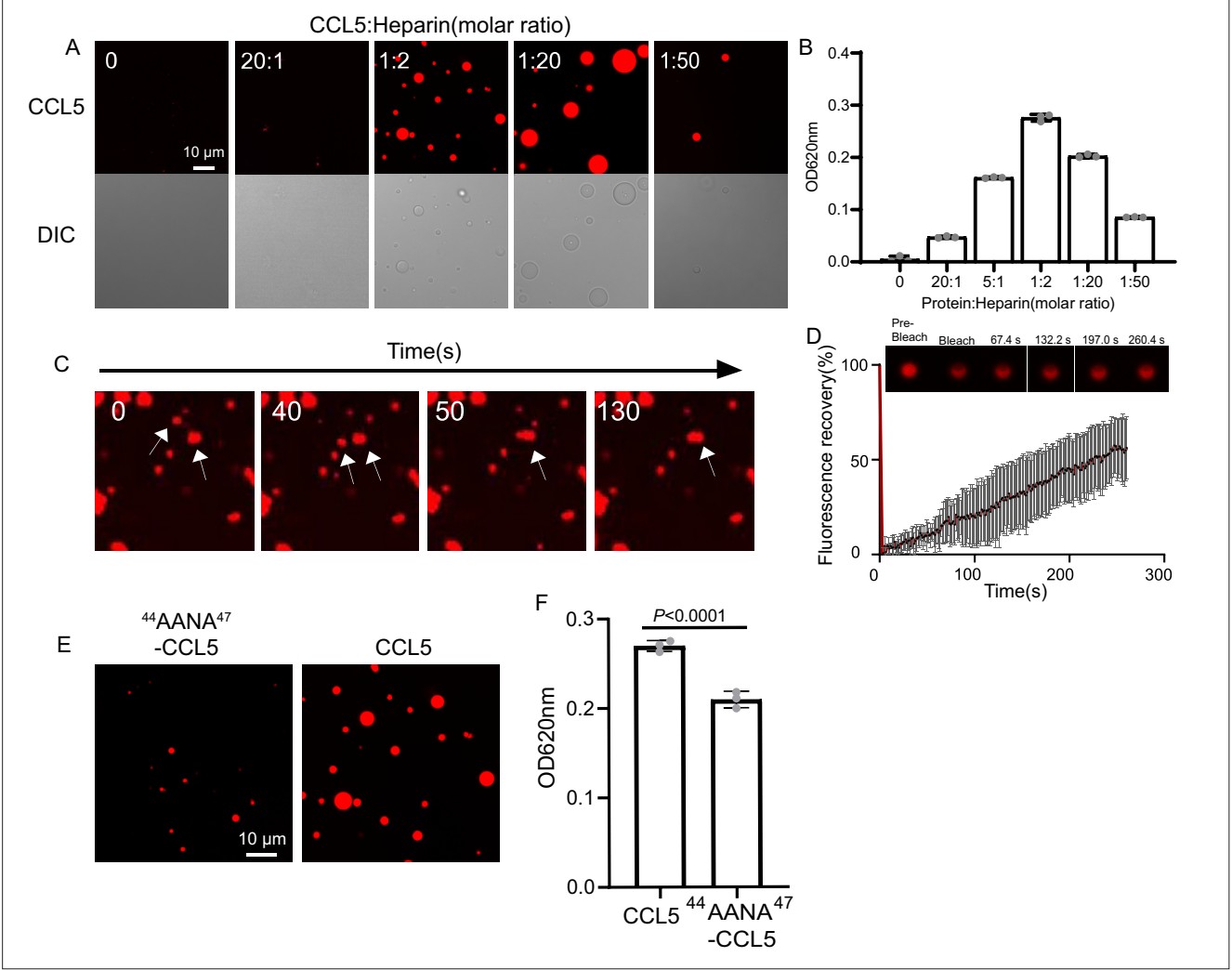

**Figure 1.** CCL5 phase separates with heparin in solution. (**A**) Confocal images of assembling status of 20 μM CCL5-Cy3 mixed with heparin at different ratios and 5% PEG (Polyethylene Glycol). Scale bar = 10 μm. (**B**) Turbidity changes with increase of heparin concentration were measured at 620 nm. (**C**) Fusion of phase-separated droplets formed at the ratio of CCL5-Cy3:heparin = 1:2. The white arrows indicate the dynamic fusion of two adjacent droplets. (**D**) Representative fluorescence recovery after photobleaching (FRAP) results of droplets formed by CCL5-Cy3:heparin = 1:2 depicted in **A**, showing the intensity of fluorescence pre- and after photobleachin. The images of representative droplets in different recovery stages are shown. (**E**) Confocal images of assembling status of [44]AANA[47]-CCL5 or CCL5 in the presence of heparin. Scale bar = 10 μm. (**F**) Comparison of CCL5 and [44]AANA[47]-CCL5 turbidity in the presence of heparin. Data are mean ± standard deviation (s.d.). $n = 3$ (for **B, F**). Normal distribution was assessed by the Shapiro–Wilk (SW) normality test. p values were determined by unpaired two-tailed $t$-tests.

The online version of this article includes the following source data and figure supplement(s) for figure 1:

**Source data 1.** Excel file containing the results of turbidity and descriptive statistics in *Figure 1B*.

**Source data 2.** Excel file containing relative fluorescent value in *Figure 1D*.

**Source data 3.** Excel file containing the results of turbidity and descriptive statistics in *Figure 1F*.

**Figure supplement 1.** Protein expression and purification.

**Figure supplement 1—source data 1.** Original file for the sodium dodecyl sulfate–polyacrylamide gel electrophoresis (SDS–PAGE) in *Figure 1—figure supplement 1A*.

**Figure supplement 1—source data 2.** Image containing *Figure 1—figure supplement 1A* and original scans of the sodium dodecyl sulfate–polyacrylamide gel electrophoresis (SDS–PAGE) with sample labels.

**Figure supplement 1—source data 3.** Original file for the sodium dodecyl sulfate–polyacrylamide gel electrophoresis (SDS–PAGE) in *Figure 1—figure supplement 1B*.

**Figure supplement 1—source data 4.** Image containing *Figure 1—figure supplement 1B* and original scans of the sodium dodecyl sulfate–polyacrylamide gel electrophoresis (SDS–PAGE) with sample labels.

*Figure 1 continued on next page*

**Figure supplement 2.** The effect of salt concentration on phase separation.

**Figure supplement 3.** Comparison of binding of CCL5 and [44]AANA[47]-CCL5 to heparin tetrasaccharide.

molecular docking analysis using a tetrasaccharide structure of heparin (*Figure 1—figure supplement 3A*). The results showed the docking of heparin tetrasaccharide into human CCL5, forming crucial electrostatic contacts at the CCL5 dimer interface, where it packed tightly against Arg 44, Lys45, and Arg47 (*Figure 1—figure supplement 3A*). In comparison, binding free energy of heparin to the [44]AANA[47]-CCL5 mutant was significantly higher (*Figure 1—figure supplement 3B*).

## Heparin-dependent phase separation is one essential step for CCL5 gradient formation

To elucidate the intrinsic connection between the heparin-dependent phase separation of CCL5 and its chemotactic activity, we developed an in vitro diffusion assay in which purified His-CCL5-EGFP (400 ng/µl) (*Figure 1—figure supplement 1B*) was immobilized on heparin-beads or on Ni-NTA-beads. In 96-well plates, the beads were embedded in the center of Matrigel (*Makarenkova et al., 2009*) and incubated for 12 hr. Diffusion of the His-CCL5-EGFP was monitored by measuring fluorescence intensity along a line interval passing through the bead localized center. As shown in *Figure 2A*, His-CCL5-EGFP diffused a considerable distance away from the heparin-beads, forming a long and shallow gradient, whereas His-CCL5-EGFP did not diffuse from the Ni-NTA beads (*Figure 2B*). In comparison, [44]AANA[47]-CCL5 binding to the heparin-beads was significantly weaker and did not diffuse (*Figure 2A*, right panel). To exclude the potential effect of EGFP, we repeated the experiment using Cy3-tagged CCL5. In a similar manner, the heparin-beads bound CCL5-Cy3 diffused in Matrigel but not the Ni-NTA beads bound CCL5-Cy3 as revealed by three-dimensional (3D) imaging (*Figure 2—figure supplement 1* and *Figure 2—figure supplement 2*). These results suggest that heparin-beads tethered CCL5 via phase separation, which enables rapid exchange with the external environment leading to diffusion and gradient formation; while the Ni-NTA beads bound CCL5 lacks phase separation, therefore no diffusion.

Next, to test whether the formed chemokine gradients of CCL5-heparin-beads contribute to the chemotactic activity, we established an in vitro chemotaxis assay using Transwell (*Proudfoot et al., 2003*) in which CCL5 either in heparin-beads or in Ni-beads were placed in the lower chamber (*Figure 2C*) and THP-1 cells were placed in the upper chamber. Again, CCL5-heparin-beads showed robustly higher chemotactic activity (*Figure 2D*), while CCL5-Ni-NTA beads essentially did not induce chemotaxis. In comparison, [44]AANA[47]-CCL5-heparin-beads showed relatively weaker chemotactic activity than WT-CCL5-heparin-beads (*Figure 2D*). The dramatic difference in the chemotactic activity of CCL5 between immobilization on heparin-beads or Ni-NTA beads indicates that the chemotactic function is achieved by establishment of a functional gradient rather than just immobilization.

## HS-dependent CCL5 phase separation and its chemotactic activity

Having seen the functional roles of heparin in CCL5 gradient formation in solution and Matrigel, we wanted to find out whether HS on the cell surface also can phase separate with CCL5 using a pair of well-established cell lines, CHO-K1 (wild-type) and CHO-677 (mutant lacking HS). Co-incubation of CCL5-Cy3 with the cells resulted in strong signals on the cell surface of CHO-K1 revealed by confocal microscopy, indicating formation of puncta surrounding the cells (*Figure 3A*). However, CHO-677 completely lacked the signals. The finding was further verified by Z-stack scanning (*Figure 3B*). Fluorescence recovery of about 40% condensates after photobleaching (FRAP) indicates there is a liquidity of the condensates on CHO-K1 (*Figure 3C*). These results indicated that CCL5 formed phase separation with HS on the CHO K1 cell surface, which was not possible in CHO-677 cells lacking HS.

In order to know whether the HS-dependent CCL5 phase separation is essential for the chemotactic activity of CCL5, we established a cell-based chemotaxis assay (*Figure 3D*). CHO-K1 cells along with CCL5 was placed in the lower chamber for 1 hr and THP-1 cells was seeded in the upper chamber, separated by a porous membrane. After incubation for 4 hr, THP-1 cells transmigrated into the lower chamber was quantified. The results revealed that both CCL5 alone and CHO-K1 alone had low chemotactic activity; in contrast, CHO-K1 in the presence of CCL5 showed significantly higher chemotactic

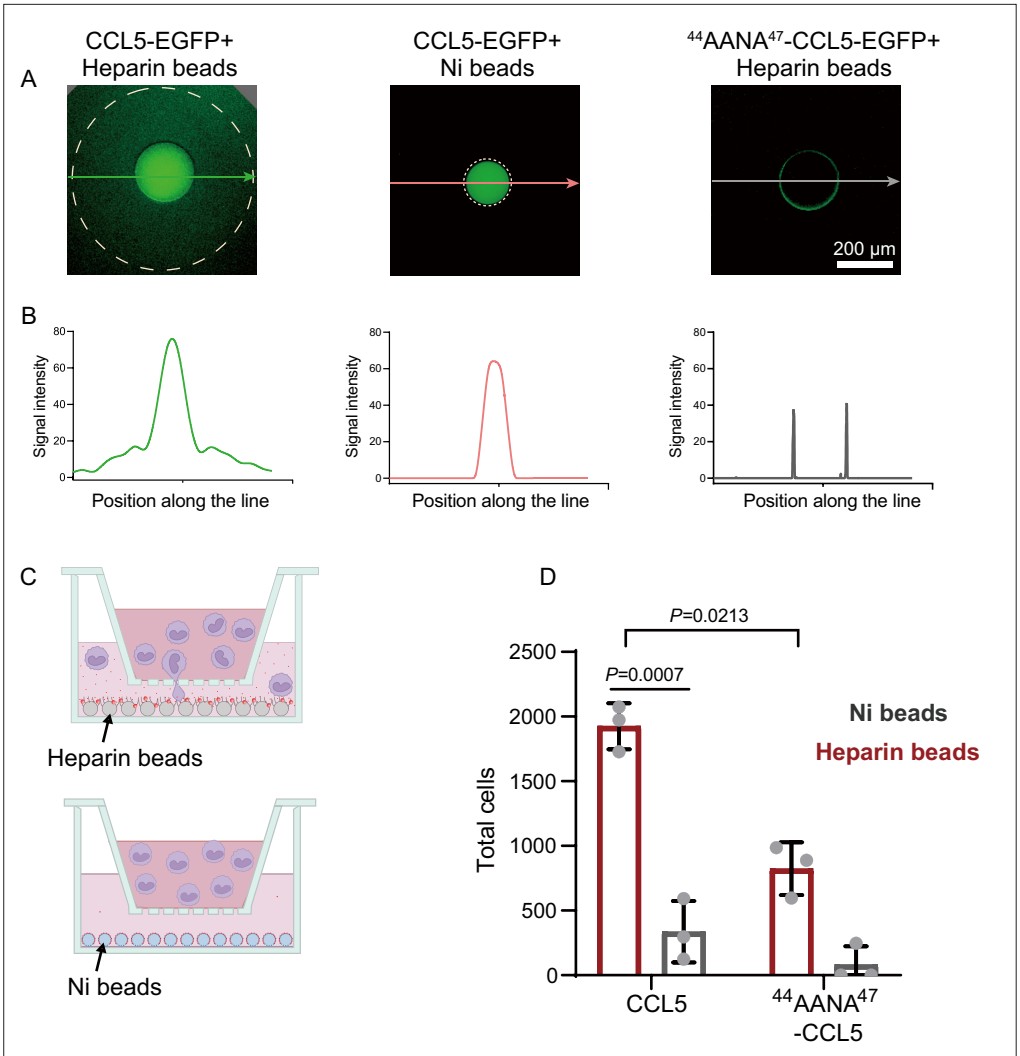

**Figure 2.** Co-phase separation of CCL5 and heparin establishes chemokine gradient. (**A**) CCL5-EGFP or [44]AANA[47]-CCL5-EGFP were bound to heparin-beads or Ni-NTA beads, respectively, and were placed in Matrigel in 96-well plate. After incubation for 12 hr images were taken to quantify the fluorescence intensity. (**B**) Quantification of the fluorescence signals along the lines with arrows indicated in **A**. (**C**) Illustration of in vitro chemotaxis assay. (**D**) Heparin-beads or Ni-NTA beads bound with CCL5 or [44]AANA[47]-CCL5 were placed in the lower chamber. THP-1 cells ($3 \times 10^5$ cells) were added to upper chamber. After 4 hr, THP-1 in the lower chamber was collected and counte. Data are mean ± standard deviation (s.d.). $n$ = 3. Normal distribution was assessed by the Shapiro–Wilk (SW) normality test. p values were determined by unpaired two-tailed $t$-tests.

The online version of this article includes the following source data and figure supplement(s) for figure 2:

**Source data 1.** Excel file containing output results of gray value in *Figure 2B*.

**Source data 2.** Excel file containing the results of cell counting and descriptive statistics in *Figure 2D*.

**Figure supplement 1.** Chemotactic function of CCL5-EGFP.

**Figure supplement 2.** Diffusion of the CCL5-Cy3 in heparin-beads or Ni-NTA beads.

activity (*Figure 3E*). In the same line, the chemotactic activity of CCL5 was much lower when incubated with CHO-677 in the lower chamber. As it was found that higher proportion of heparin reduced phase separation of CCL5 (*Figure 1A*), we tested whether this is the case in cells. Indeed, addition of heparin in the incubation of CHO-K1 abolished the phase separation of CCL5-Cy3 on the cells surface (*Figure 3A*, lower panel), accordingly attenuated the chemotactic activity of CCL5 (*Figure 3E*). When [44]AANA[47]-CCL5 was incubated with CHO-K1 cell, no obvious phase-separated condensates appeared on the cell surface (*Figure 3—figure supplement 1*), and consequently, [44]AANA[47]-CCL5 showed

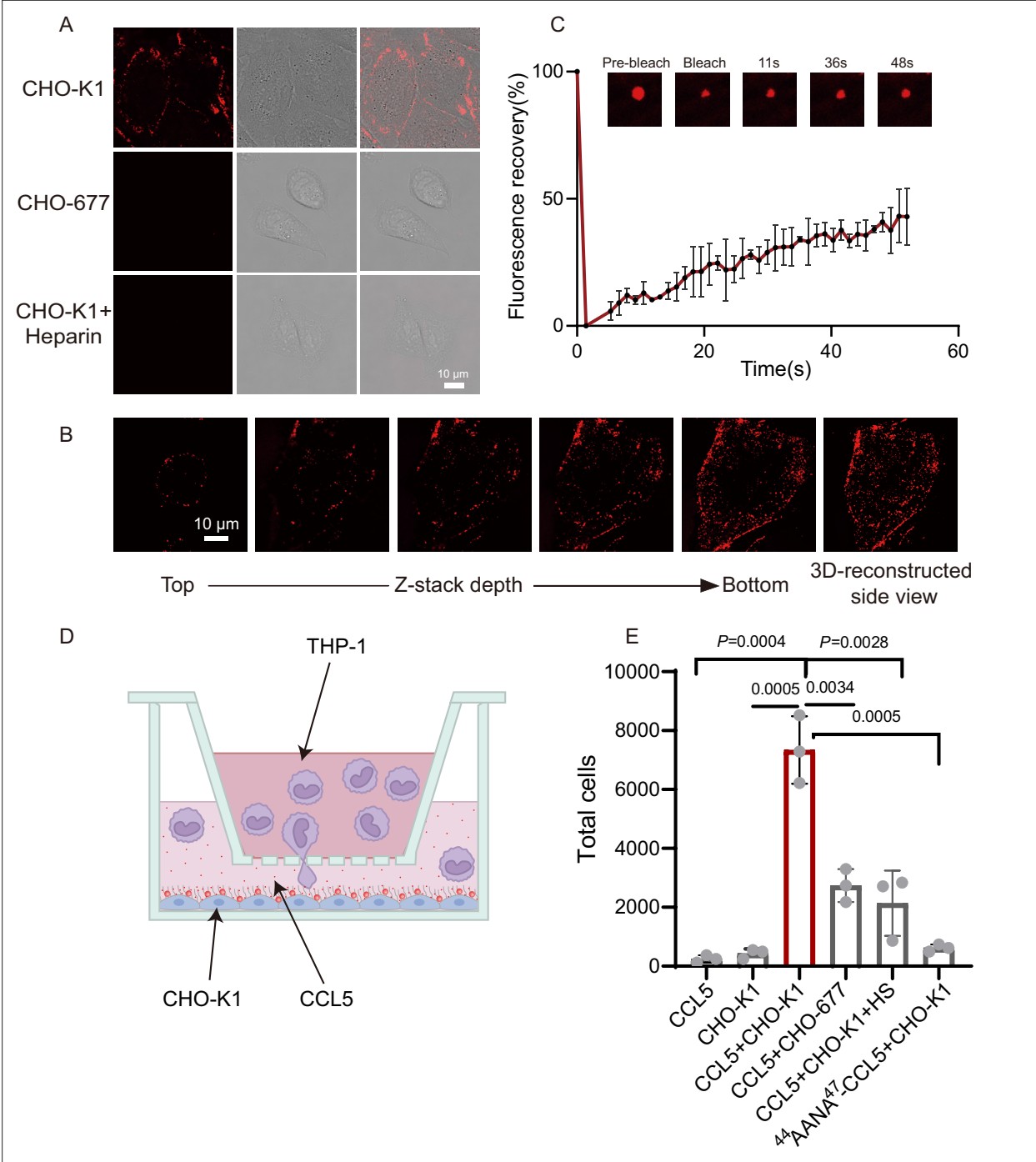

**Figure 3.** CCL5 phase separates with heparan sulfate on the cell surface. (**A**) Microscopy images of CCL5-Cy3 on the surface CHO-K1 and CHO-677 as well as CHO-K1 treated with 1 mg/ml heparin. From left to right, the images are fluorescent-field, bright-field, and overlay of two illuminations. Scale bar = 10 μm. (**B**) Z-stack scanning of CCL5-Cy3 phase separation on CHO-K1 cell surface. The cell was imaged by confocal microscope with the Z-stack method. Scale bar = 10 μm. (**C**) Fluorescence recovery after photobleaching (FRAP) of the condensates formed by CCL5-Cy3, showing the intensity of fluorescence pre- and after photobleaching. The size of the representative droplets in different recovery stages are shown above of the graph. (**D**) Graphical illustration of the cell-based chemotaxis assay. CHO cells ($1 \times 10^5$ cells/well) were plated onto the lower chamber for 24 hr (fully attached) and CCL5 or [44]AANA[47]-CCL5 or heparin were added as indicated. THP-1 cells ($3 \times 10^5$) were placed on upper chambers. After 4 hr, a small volume of medium in the lower chamber was aspirated to count THP-1 transmigrated through the membrane. (**E**) Quantification of THP-1 collected from the lower chambe. Data are mean ± standard deviation (s.d.). $n$ = 3. Normal distribution was assessed by the Shapiro–Wilk (SW) normality test. p values were determined by unpaired two-tailed $t$-tests.

The online version of this article includes the following source data and figure supplement(s) for figure 3:

*Figure 3 continued on next page*

*Figure 3 continued*

**Source data 1.** Excel file containing relative fluorescent value in *Figure 3C*.

**Source data 2.** Excel file containing the results of cell counting and descriptive statistics in *Figure 3E*.

**Figure supplement 1.** $^{44}$AANA$^{47}$-CCL5 did not immobilize to the surface of CHO-K1.

reduced chemotactic activity when incubated with CHO-K1 (*Figure 3E*) in comparison with WT CCL5. Thus, we may conclude that HS is essential on the CHO cell surface for phase separation of CCL5 and its chemotaxis function.

Based on the observation that CCL5 forms a concentration gradient on heparin-beads where heparin-driven phase separation enabled CCL5 diffuse, we hypothesized a similar scenario of HS on cells, for example, CCL5 was immobilized or condensed by HS on the cell surface through phase separation, and gradually diffuse to form a gradient to guide cell migration. To test this conjecture, we transfected the construct of CCL5-EGFP into CHO-K1. The cells were co-cultured with human umbilical vein endothelial cells (HUVECs) (*Øynebråten et al., 2015*). Released CCL5-EGFP from CHO-K1 cells was readily taken up by nearby HUVEC cells, formed bright phase-separated condensates on the cell surface (*Figure 4A* and *Figure 4—figure supplement 1A*). With extended incubation, the condensates became fused (*Figure 4B*), in a similar way of CCL5-heparin condensates in solution (*Figure 1C*). The FRAP experiment showed reasonable FRAP, indicating liquid-like properties of the condensates on the cell surface (*Figure 4C*). To further illustrate the gradient formation of the CCL5-EGFP condensates, we cultured HUVEC in Matrigel and embedded tiny amounts (about 400 cells) of the CCL5-EGFP transfected CHO-K1 as a source of CCL5-EGFP. Confocal microscopy monitoring found decreased concentration of CCL5-EGFP with increasing distance of HUVEC from the source cells (CCL5-EGFP expressing CHO), confirming that CCL5-EGFP established a gradient by phase separation on its target cells (*Figure 4D, E*, *Figure 4—figure supplement 1B, C*). Further, both CHO-K1 and CHO-677 transfected with CCL5-EGFP were analyzed on the same setting. The results show that a gradient was formed in CHO-K1 cells when co-cultured with CCL5-EGFP transfected CHO-K1 in Matrigel, while CHO-677 failed to form the concentration gradient (*Figure 4—figure supplement 1D*). Furthermore, the chemotactic activity was demonstrated by placing CCL5-EGFP transfected CHO-K1 in a lower chamber and THP-1 in the upper chamber. Counting the transmigrated THP-1 cells showed that the transfected CHO-K1 had stronger chemotactic activity compared with WT-CHO-K1 (*Figure 4F*).

## Ex vivo and in vivo demonstration of HS-dependent chemotactic activity of CCL5

To further verify our findings, we placed HUVEC or CHO cells in the lower chamber in the presence of CCL5 and blood cells that lysed red cells in the upper chamber of the Transwell device to detect transmigration of the inflammatory cells. Quantification of the transmigrated cells into the lower chamber after incubation for 4 hr revealed that neither CCL5 alone nor CHO-677 in the presence of CCL5 showed substantial chemotactic activity; in contrast, HUVEC and CHO-K1 in the presence of CCL5 showed significantly higher chemotactic activity. It should be pointed out that the total blood cells that lysed red cells in the three blood samples varied dramatically, resulting in the great deviations in the number of transmigrated cells of each sample. Nevertheless, the trend of chemotactic activity under the different conditions is consistent in all samples (*Figure 5*).

To verify our findings in vivo, wild-type, or mutant CCL5 was injected into the peritoneal cavity of Balb/c mice and peritoneal cells recruitment was monitored (*Proudfoot et al., 2003*). Wild-type CCL5 induced a robust increase of total cells in the peritoneal lavage to a level approximately four-fold over the saline control (*Figure 6*). In comparison, though $^{44}$AANA$^{47}$-CCL5 group had a higher cell number than saline control, the recruited cells were much less than CCL5. Heparin (1 mg/ml) co-injected with wild-type CCL5 attenuated the recruitment of inflammatory cells, which, again, can be ascribed to the competitive interaction with endogenous HS in binding to CCL5. Overall, these results demonstrate that the phase separation of CCL5 with HS indeed contribute to chemotaxis function of CCL5 in vivo.

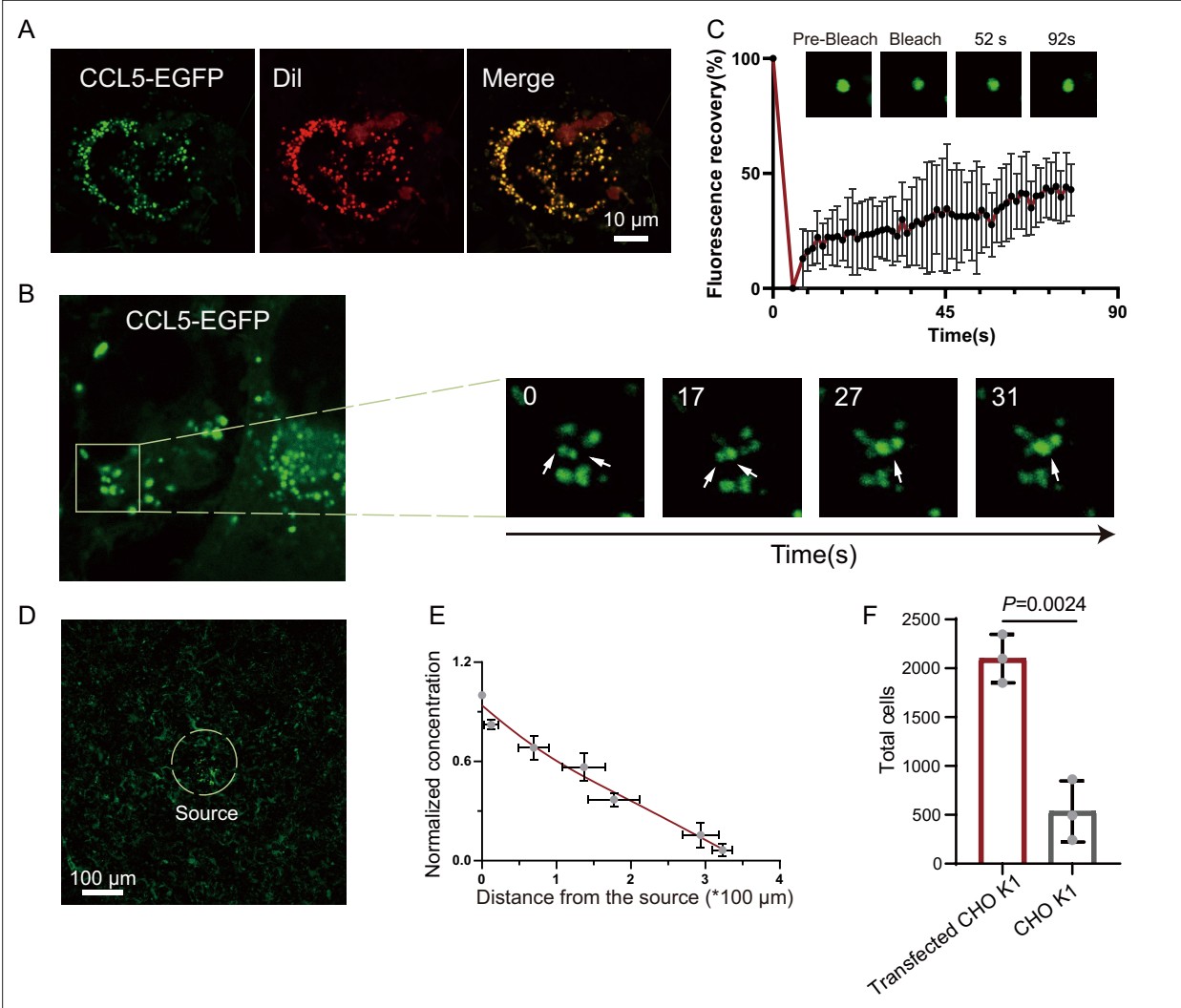

**Figure 4.** Formation of chemokine gradient on the cell surface by phase separation. (**A**) Human umbilical vein endothelial cells (HUVECs) were seeded on the plates for 24 hr and stained with Dil (Cell membrane red fluorescent probe). After washing with PBS (Phosphate buffer saline), CCL5-EGFP transfected CHO-K1 were added and co-cultured for 24 hr before taking the images (scale bar = 10 μm). (**B**) Fusion events of droplets formed by CCL5-EGFP on HUVEC cell surface. The white arrows in cropped images indicate the fusion of two droplets with time. (**C**) Fluorescence recovery after photobleaching (FRAP) of the condensates formed by CCL5-EGFP in HUVEC cell surface, showing the intensity of fluorescence pre- and after photobleaching. The images of representative droplets in different recovery stages are shown. (**D**) HUVEC cells were seeded on the plate and firmly adhered before adding the 400 CCL5-EGFP transfected CHO-K1 cells placed in 50% of Matrigel. After 1-hr co-culture, confocal images were taking showing CCL5-EGFP diffusion on the surface of HUVEC. Dotted white circle indicates the source cell of CCL5-EGFP transfected CHO-K1. Scale bar = 100 μm. (**E**) Quantification of the fluorescence intensity shows decreased signals of CCL5-EGFP as the distance from the source cells increased. (**F**) The chemotaxis assay (same experimental conditions as described in *Figure 1*) shows higher activity of CCL5-EGFP transfected CHO-K1 cells than wild-type CHO-K1 cell. Data are mean ± standard deviation (s.d.). *n* = 3. Normal distribution was assessed by the Shapiro–Wilk (SW) normality test. p values were determined by unpaired two-tailed *t*-tests.

The online version of this article includes the following source data and figure supplement(s) for figure 4:

**Source data 1.** Excel file containing relative fluorescent value in *Figure 4C*.

**Source data 2.** Excel file containing output results of gray value and normalized concentration in *Figure 4E*.

**Source data 3.** Excel file containing the results of cell counting and descriptive statistics in *Figure 4F*.

**Figure supplement 1.** Diffusion of CCL5-EGFP on the cell surface.

Biochemistry and Chemical Biology | Cell Biology

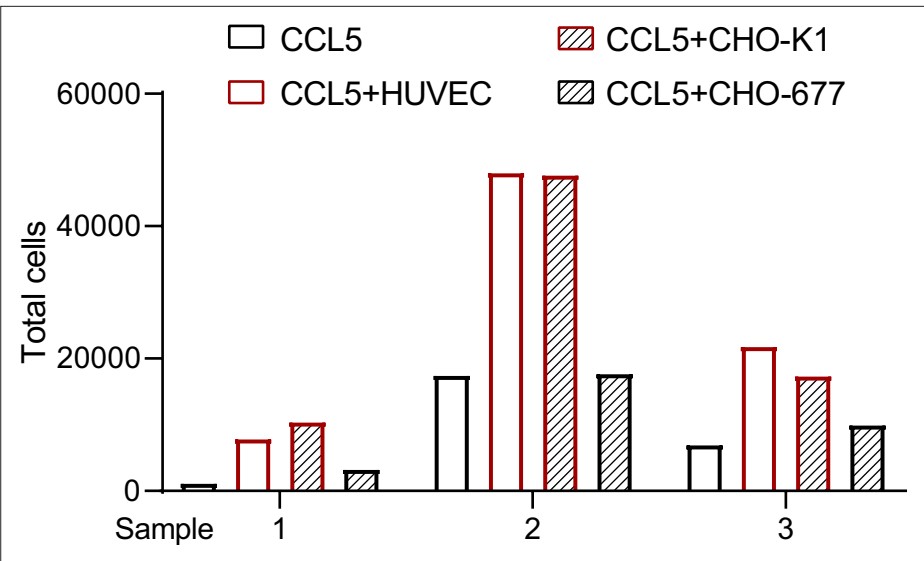

**Figure 5.** Transmigration of blood cells that lysed red cells. The cells (CHO-K1, CHO-677, or human umbilical vein endothelial cell [HUVEC]) (1 × 10⁵ cells/well) as indicated were plated onto the lower chamber for 24 hr (fully attached) and CCL5 was added for 1 hr. The blood cells that lysed red cells isolated from three healthy volunteer donors were placed in the upper chamber (total number of cells: Sample 1: $8.65 \times 10^5$; Sample 2: $2.31 \times 10^6$; Sample 3: $2.23 \times 10^6$). After culturing for 4 hr, the total transmigrated cells in the lower chamber were counted.

The online version of this article includes the following source data for figure 5:

**Source data 1.** Excel file containing the results of cell counting in *Figure 5*.

## Discussion

It has been suggested that an important component of the chemokine signaling is formation of a haptotactic gradient through immobilization of chemokines on cell surface GAGs (*Makarenkova et al., 2009*). Interactions with GAGs facilitate gradient formation of chemokine, providing directional cues for migrating cells. HS is one member of the GAGs family and ubiquitously expressed on the cells surface of endothelial and epithelial cells, forming glycocalyx (*Simon Davis and Parish, 2013*). It is known that HS (and its analog heparin) binds to a broad spectrum of cytokines, functioning as a co-receptor to mediate signaling activity of the cytokines (*Xie and Li, 2019*). Several inflammatory chemokines, including CCL5, bind to HS/heparin (*Proudfoot et al., 2003*); however, the molecular mechanisms of HS in CCL5-induced chemotactic activity have not been reported.

Phase separation is a common mechanism for protein assembly and compartmentalization, and it contributes to a variety of cellular processes, including the formation of membraneless organelles, signaling complexes, the cytoskeleton, and numerous other supramolecular assemblies (*Hyman et al., 2014*; *Zbinden et al., 2020*). Emerging evidence indicates that biomolecules in phase-separated liquid droplets are mobile and transitorily interact with surrounding molecules (*Nakashima et al., 2019*). Most of CC chemokines are found to occur oligomerization in vivo, which may be modulated by GAGs (*Hoogewerf et al., 1997*). However, the exceptional species of CCL7 is present at monomer, and has multiple GAG-binding epitopes, enabling it to functions as a non-oligomerizing chemokine (*Salanga et al., 2014*). In comparison, CXCL4 (Platelet Factor 4) binds strongly to heparin (*Shi et al., 2023*) and HS (*Horton et al., 2021*), leading to formation of oligomers and aggregates. It should be interesting to examine whether these chemokines also undergo phase separation and to find out whether LLPS is controlled by oligomerization.

Earlier study reported that the conserved and positively charged BBXB motif in the CC chemokines is the key to interact with negatively charged GAGs through multivalent weak ionic interactions (*Liang et al., 2016*). Structural study revealed that the basic amino acids R44, K45, and R47 create two positively charged regions on the interface of CCL5 dimer, facilitating the interaction with heparin (*Lortat-Jacob et al., 2002*). Based on above findings, using our established LLPS method, we demonstrated that CCL5 phases separate on the cell surface via an HS-dependent mechanism, which is required for

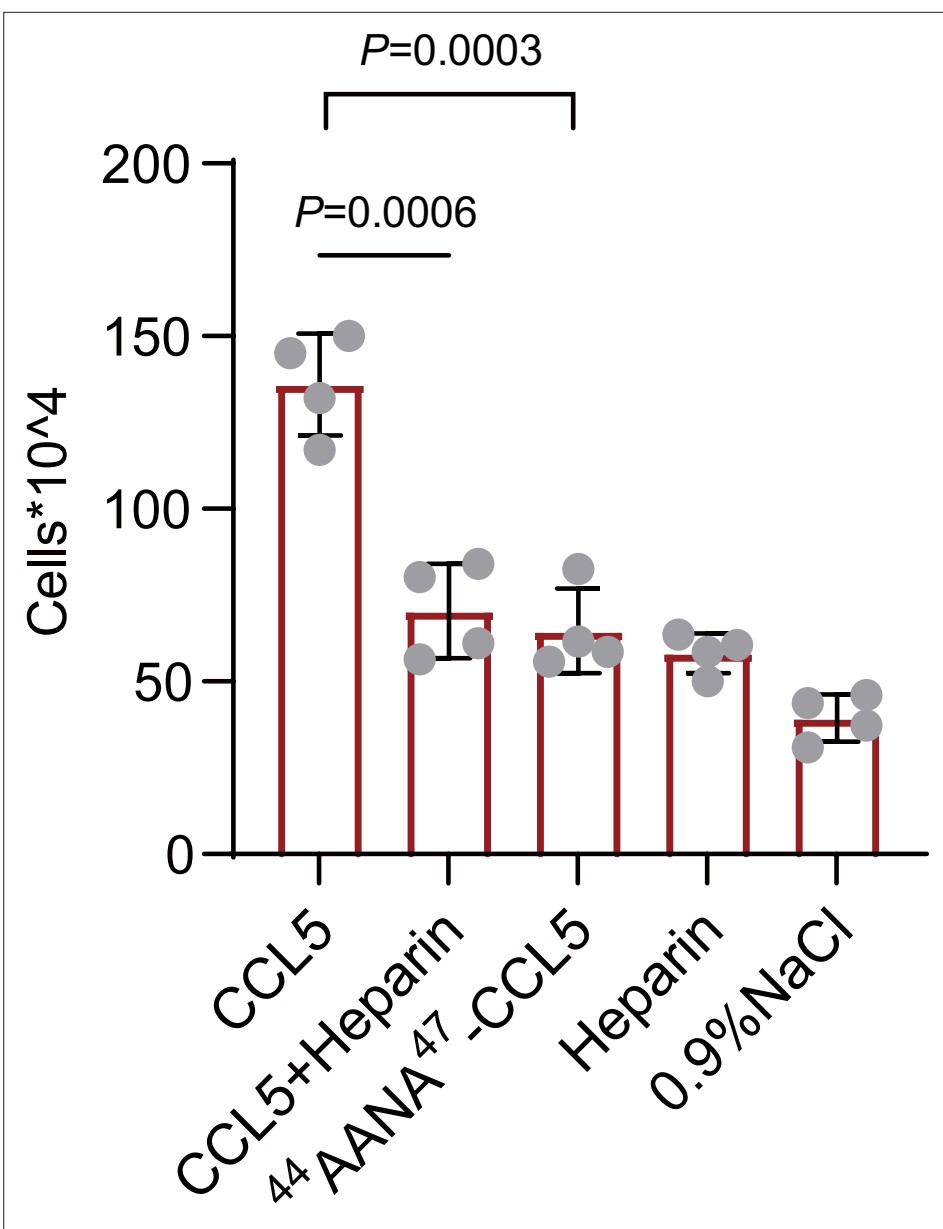

**Figure 6.** Chemokines phase separation promotes cell recruitment in vivo. Mice were treated by intraperitoneal injection of CCL5 or the reagents as indicated. After 18 hr, the animals were sacrificed and peritoneal lavage was collected. Total cell number was counted by automated cell counter. Data are mean ± standard deviation (s.d.). $n$ = 4. Normal distribution was assessed by the Shapiro–Wilk (SW) normality test. p values were determined by unpaired two-tailed $t$-tests.

The online version of this article includes the following source data for figure 6:

**Source data 1.** Excel file containing the results of cell counting and descriptive statistics in **Figure 6**.

CCL5 chemotaxis. Mutations on the basic amino acids R44, K45, and R47 resulted in loosing binding of CCL5 to heparin-beads, accordingly, lost its chemotactic activity. This finding also supports the fact that the phase separation of CCL5 with HS is mediated by weak electrostatic interactions, which facilitates the diffusion of CCL5 in solution to form a gradient.

It is well established that HS constitutes the major component of the glycocalyx on the endothelial cell surface (**Oshima et al., 2021**), and our earlier results showed that degradation of endothelial surface HS by heparanase impaired the function of MIP-2/CXCL2-induced leukocyte rolling, adhesion, and transmigration (**Massena et al., 2010**). From the finding of this study that the ratio of

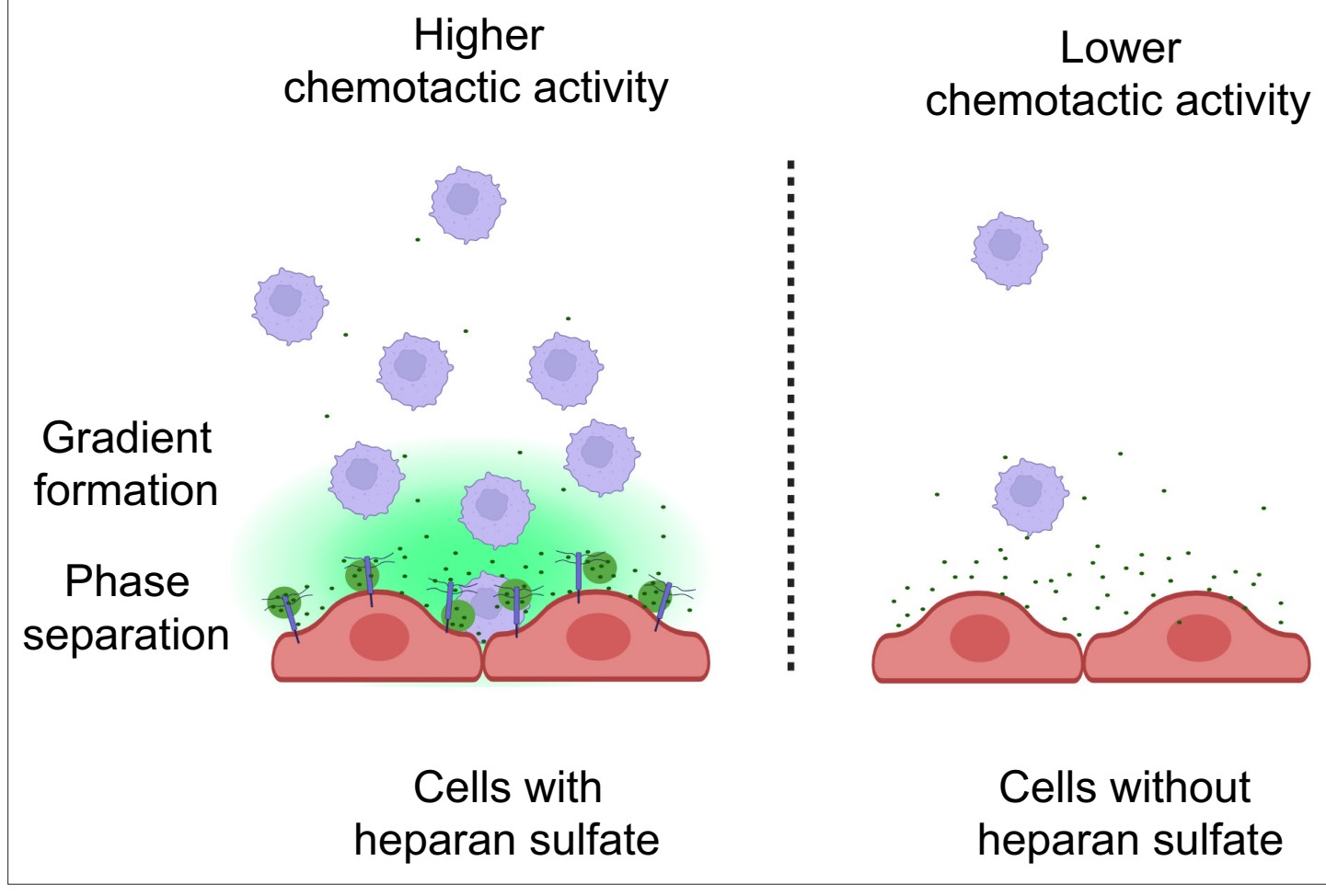

**Figure 7.** Schematic model of CCL5 phases separation along with heparan sulfate on the cell surface. The inflammatory cells secrete chemokines that are immobilized by heparan sulfate in the form of liquid–liquid phase separation and subsequently diffused to form a concentration gradient.

CCL5:heparin modulated droplets and condensates formation (*Figure 1A, B*), we may assume that HS on the cell surface should have a sufficient capacity to interact with chemokines, controlling their chemotactic activity and inflammatory cell migration. This assumption is supported by the finding that overexpressing CCL5 in CHO-K1 cells led to higher chemotactic activity, which may represent a scenario of acute inflammatory reaction.

Collectively, our studies provide the first evidence that CCL5 gradient formation and chemotactic activity are dependent on HS. The interaction between CCL5 and HS is through a weak multivalent electrostatic binding, which facilitates LLPS and diffusion of the chemokine (*Figure 7*). Notably, the results suggest that there may exist a physiological balance between CCL5 and HS, modulating the chemokine activity. This finding supports the hypothesis to develop chemokine-binding HS mimetics that may competitively interfere with the HS–chemokine binding and phase separation, accordingly modulating inflammatory reactions.

# Materials and methods

**Key resources table**

| Reagent type (species) or resource | Designation | Source or reference | Identifiers | Additional information |
|---|---|---|---|---|
| Gene (*Homo sapiens*) | CCL5 | GenBank | HGNC:10632 | |
| Strain, strain background (*Escherichia coli*) | Rosetta(DE3) | Biomed | BC204-01 | |
| Strain, strain background (*Escherichia coli*) | OrigamiB(DE3) | Biomed | BC205-01 | |
| Sequence-based reagent | [44]AANA[47]-CCL5-mut-F | ***Øynebråten et al., 2015*** | PCR primers | GCAGTCGTCTTTGTCACCGCGGC GAACGCGCAAGTGTGTGCCAACCCA |
| Sequence-based reagent | [44]AANA[47]-CCL5-mut-R | ***Øynebråten et al., 2015*** | PCR primers | TGGGTTGGCACACACTTGCGCGTTC GCCGCGGTGACAAAGACGACTGC |
| Recombinant DNA reagent | pET-28a | Miao Ling Plasmid | P51040 | |
| Chemical compound, drug | Cy3-NHS | ATT Bioquest | 1023 | |
| Chemical compound, drug | Heparin | GlycoNovo | C-HEPPIM | Mw: 13.980 Da |
| Cell line (*Homo sapiens*) | HUVEC | Pricella | CL-0675 | |
| Cell line (*Cricetulus griseus*) | CHO-K1 | GlycoNovo | | |
| Cell line (*Cricetulus griseus*) | CHO-677 | GlycoNovo | | |
| Biological sample (*Mus musculus*) | BALB/cAnNCrl | Vital River | 211 | Female |
| Biological sample (*Homo sapiens*) | Human peripheral blood cell | China-Japan Friendship Hospital | | Freshly isolated from volunteers |
| Software, algorithm | CHARMM-GUI | CHARMM-GUI | RRID:SCR_025037 | |
| Software, algorithm | GROMACS | GROMACS | RRID:SCR_014565 | |
| Software, algorithm | gmx_MMPBSA | GitHub | RRID:SCR_002630 | |
| Software, algorithm | LAS X software 3.5 | Leica Application Suite X | RRID:SCR_013673 | |

## Protein expression and purification

DNA sequences of wild-type human CCL5 (***Donlon et al., 1990***) were synthesized by General Biosystems, China. For the mutant of CCL5 ([44]AANA[47]-CCL5), site-directed mutations were introduced using the Quikchange mutagenesis kit (Beyotime Biotechnology, China) using mutagenesis primes: 5'-GCAGTCGTCTTTGTCACCGCGGCGAACGCGCAAGTGTGTGCCAACCCA-3'. CCL5 gene and [44]AANA[47]-CCL5 gene were constructed in pET-28a vector (Miao Ling Plasmid, China) and transformed into *E. coli* Rosetta (DE3) (Biomed, China) for expression (***Proudfoot et al., 1996***; ***Proudfoot and Borlat, 2000***). The products contained an additional Met residue at the N-terminus (Met-CCL5) and His tag or EGFP-fusion protein. Bacterium were grown to optical density of 0.6 at 37°C and induced with 0.5 mM isopropyl-β-D-thiogalactopyranoside (IPTG) at 37°C for 5 hr. The cells were harvested by centrifugation (4000 rpm, 20 min at 4°C), resuspended in lysis buffer (25 mM Tris–HCl, 100 mM NaCl, pH 8.0) and disrupted by sonication. The cell lysate was centrifuged at 12,000 rpm for 35 min at 4°C. To recover the recombinant CCL5 and [44]AANA[47]-CCL5 proteins that were mainly distributed in the inclusion bodies, the pellet was collected and resuspended in denaturing buffer (100 mM Tris–HCl, 6 M guanidine–HCl, pH 8.0) and disrupted by sonication. Cell debris was removed by centrifugation (30,000 × *g*, 20 min at 4°C). The proteins in the supernatant were refolded by dialysis in to refolding buffer (0.9 M guanidine–HCl, 100 mM Tris–HCl, 5 mM methionine, 5 mM cysteine, pH 8.0). After repeated change of dialysis buster, the protein debris was removed by centrifugation (30,000 × *g*, 20 min at 4°C) and the supernatants were applied to His-Trap chelating column (GE Healthcare), washed by refolding buffer and eluted with the same buffer containing 30 mM imidazole. The purified proteins were dialyzed into 1.0% vol/vol acetic acid aqueous solution twice and finally into 0.1% vol/vol trifluoroacetic acid aqueous solution. The purity of the recombinant CCL5 was analyzed by sodium

dodecyl sulfate–polyacrylamide gel electrophoresis. After lyophilization, the CCL5 proteins were dissolved in water and stored in −80°C.

For a better folding, the CCL5-EGFP and [44]AANA[47]-CCL5-EGFP fusion proteins were expressed in *E. coli* OrigamiB(DE3) and induced by 0.5 mM IPTG at 25°C for 12 hr. The proteins were purified by the same procedure as above.

## Labeling of CCL5 with Cy3

Cy3 monosuccinimidyl ester (Cy3-NHS; ATT Bioquest, USA) was mixed with CCL5 to a final concentration of 1.5 mg/ml, pH 9 adjusted with 1 M NaHCO$_3$. The mixture was incubated with shaking at 37°C for 1 hr, and then dialyzed in a 10-kDa dialysis tube (Thermo Fisher, USA) in ddH$_2$O. These Cy3-labled CCL5 was mixed with un-labeled protein at 1:40–1:20 ratio. This dilution of the labeled protein is in consideration that N-terminus of CCL5 may be labeled with Cy3, which is known to affect the activity of the protein.

## Cell culture

HUVECs were cultured in Dulbecco's modified Eagle's medium 1× with glucose (4.5 g/l), 10% fetal bovine serum (FBS), and 1% antibiotics (penicillin/streptomycin). CHO-K1/CHO-677 cells were cultured in F12K medium containing 10% FBS and 1% antibiotics. THP-1 were cultured in RPMI1640 medium containing 10% FBS and 1% antibiotics. All cells were cultured at 37°C with 5% CO$_2$ in a humidified incubator.

## Phase separation of CCL5 on cell surface

Cells were plated onto an 8-well Lab-Tek chambered coverglass (Thermo Fisher, USA) to around 70% confluency. Before imaging, the medium was discarded, and the cells were washed with PBS twice. Then, the protein solution diluted in the culture medium (500 nM) was added to the cells and incubation 1 hr in a cell incubator. Confocal microscopy image was captured with an inverted Leica DMi8 microscope, equipped with lasers for 489 and 554 nm excitation. Images were acquired using a ×63 objective (oil immersion).

## Z-stack for living cell 3D rendering

3D reconstruction for living cells were implemented with an inverted Leica DMi8 microscope. Images were acquired using the ×63 oil immersion lens, a pinhole of 1 AU, 522 nm laser with 10% laser power, followed by setting the starting position and end position of Z-stack, 100–200 Nr. of Steps or 1 μm z-step size was selected.

## Imaging of CCL5 phase separation in vitro

CCL5 were diluted to 4–6 mg/ml in KMEI buffer (150 mM KCl, 1 mM MgCl$_2$, 1 mM Ethylenebis (oxyethylenenitrilo) tetraacetic acid (EGTA) and 10 mM imidazole, pH 6.5). For the co-phase separation of CCL5 and heparin (GlycoNovo, China, Mw: 13.980 kDa), 20 μM of CCL5 were mixed with heparin and 5% wt/vol PEG-8000 in the assay buffer. The mixtures were incubated at 37°C for 30 min then were loaded into a 96-well plate for imaging analysis. Images were captured with a Leica DMi8 confocal microscopy with a ×63 objective (oil immersion) and LAS X software 3.5.

## Turbidity assay

CCL5 proteins were mixed with various concentrations of heparin (0–1000 μM) and PEG-8000 (5% wt/vol) in the KMEI buffer. After incubation at 37°C for 30 min the mixture was transferred to a 96-well plate and turbidity was measured by absorption at 620 nm using a Multiskan FC microplate reader (Thermo Fisher, USA). All samples were examined in triplicates ($n = 3$).

## Molecular docking

Heparin tetrasaccharide was docked to CCL5 (PDB number: 5coy) wild-type and [44]AANA[47]-CCL5 (PDB number: 1u4r) mutation by ClusPro server (*Desta et al., 2020*). The docked models were selected according to the following standards: (1) close to the critical residues, that is, aa 44–47 and (2) having the lowest energy score. The selected CCL5–heparin tetrasaccharide complex was handled by Glycan Reader & Modeler module of CHARMM-GUI (*Damm et al., 1997*; *Liu et al., 2013*) to set up the

molecular dynamics simulation system and generate the input files. The energy minimization (EM) simulations of the complexes were performed with GROMACS [https://doi.org/10.1016/j.softx.2015.06.001], and based on the EM trajectory, the binding free energy between CCL5 and heparin tetra-saccharide was calculated with gmx_MMPBSA (*Van Wart et al., 2014*).

## Diffusion of CCL5 on heparin-beads and Ni-NTA beads

Heparin-beads (Solarbio, China) and Ni-NTA beads (GenScript, China) 50 µl gel were mixed with 400 ng/µl of His-CCL5-EGFP and [44]AANA[47]-CCL5-EGFP fusion proteins in a total volume of 500 µl. After incubation for 30 min in ice, the beads were dropped into Matrigel (ABWbionova, China) in 96-well plates kept on ice. Then the plates were moved to a 37°C incubator for 12 hr. Images were captured with a Zeiss Axiocam 506 color Fluorescence Microscope with a ×5, ×10 objective and analyzed with ImageJ.

## Chemotaxis assay

CHO-K1, mutant CHO-677, and HUVEC cells were cultured in the lower chamber of Transwell with 8 µm pores (Corning, USA) with a density $1 \times 10^5$ cells/well for 24 hr. Cells were washed three times with PBS and then 500 nM CCL5 or the mixed solution of 500 nM CCL5 and 1 mg/ml heparin (GlycoNovo, China) in culture medium were added and incubation 1 hr in a cell incubator prior to the chemotactic assay. Then the THP-1 cells ($3 \times 10^5$) in RPMI1640 medium were placed onto upper chamber of Transwell and incubated for 4 hr in a cell incubator, the transmigrated cells (in suspension) in the lower chamber were collected and counted by cell counter and inverted microscope.

For chemotaxis assay using beads, CCL5-EGFP and [44]AANA[47]-CCL5-EGFP were first mixed with the beads as described above, washed and diluted with PBS and then placed on the bottom of lower chamber. The THP-1 cells were placed in the upper chamber. Cell migration analysis was performed as described above.

## Diffusion of CCL5-EGFP on the cell surface

CCL5-EGFP was constructed in pCDNA3.1-EGFP vector (General Biosystems, China) and transiently expressed in CHO-K1 cells. Briefly, the cells were seeded on 6-well plated at a density of $2.5 \times 10^5$ cells/well in 2 ml medium for 24 hr and then transfected using Lipo8000 Transfection Reagent (Beyotime Biotechnology, China). After 12 hr; transfected cells were detached by pancreatin (2000 / ml). HUVEC cells were plated onto an 8-well Lab-Tek chambered coverglass (Thermo Fisher, USA) with a density $3 \times 10^4$ cells/well for 24 hr. The transfected CHO-K1 cells (2000/ml) in 200 µl medium were added to the 8-well coverglass coating HUVEC. After 1 hr, medium was aspirated and replaced with 50% of Matrigel. The transfected CHO-K1 was co-cultured with HUVEC cells for 12 hr, and images were captured with Leica DMi8 confocal microscopy with a ×20 objective. For the capture of droplets on cell surface and FRAP, the cellular membrane of HUVEC was stained with Dil (Cell membrane red fluorescent probe, Beyotime, China) for 20 min and washed three times with PBS. Images were captured with a ×100 objective (oil immersion).

## Human blood cells transmigration

Blood (1 ml) was collected from three healthy volunteers (we did not collect demographics) in EDTA (Ethylene Diamine Tetraacetic Acid) tubes and centrifuged at $500 \times g$ (4°C for 5 min). After aspiration of plasma, the cells were treated with 10 ml of Red Blood Cell Lysis solution (Beyotime Biotechnology, China) for 10 min at 4°C and centrifuged. After removing supernatant, the cells were washed three times with PBS, and resuspended in 400 µl of RPMI1640 medium and counted. The cells suspension (100 µl) was added to the upper chamber for the chemotaxis assay as described above. All blood samples were obtained with informed consent, and the study was approved by the ethics review committee of China-Japan Friendship Hospital (2022-KY-050). All relevant ethical regulations of China-Japan Friendship Hospital and governmental regulations were followed.

## Recruitment of inflammatory cells in peritoneal lavage

Eight-week-old female BALB/c mice were purchased from Charles River and kept in a pathogen-free environment with fed ad lib. The procedures for care and use of animals were approved by Beijing Municipal Science & Technology Commission, Administrative Commission of Zhongguancun Science

Park (SYXK-2021-0056) and all applicable institutional and governmental regulations concerning the ethical use of animals were followed. Mice were injected intraperitoneally with 200 μl of NaCl (0.9%, lipopolysaccharide-free), 1 mg/ml heparin, 500 nM wild-type CCL5, 500 nM $^{44}$AANA$^{47}$-CCL5 or 500 nM wild-type CCL5 added by 1 mg/ml heparin diluted into 200 μl of NaCl (0.9%, lipopolysaccharide-free), respectively. At 18-hr post-injection mice were sacrificed by cervical dislocation, the cells in the peritoneal cavity were collected by lavage of 5 ml ice-cold PBS. The total cells collected were counted with automated cell counter (Bodboge, China).

## Acknowledgements

We thank Dr. Shuibing Chen and Dr. Yongxiang Chen for help editing the manuscript and the support from Beijing Advanced Innovation Centre for Soft Matter Science and Engineering is acknowledged. National Key R&D Program of China 2021YFC2103900 to SZL; National Natural Science Foundation of China 22261132513, 22277009, 22077010 to SZL; Joint Project of BRC-BC (Biomedical Translational Engineering Research Center of BUCT-CJFH) (XK2023-14, XK2022-07) to SZL; Swedish Research Council (2021-01094) to JPL; Postdoctoral Fellowship Program of CPSF (GZC20230211) to PFP.

## Additional information

### Funding

| Funder | Grant reference number | Author |
|---|---|---|
| National Key R&D Program of China | 2021YFC2103900 | Shi-Zhong Luo |
| National Natural Science Foundation of China | 22261132513 | Shi-Zhong Luo |
| National Natural Science Foundation of China | 22277009 | Shi-Zhong Luo |
| National Natural Science Foundation of China | 22077010 | Shi-Zhong Luo |
| Joint Project of BRC-BC (Biomedical Translational Engineering Research Center of BUCT-CJFH) | XK2023-14 | Shi-Zhong Luo |
| Joint Project of BRC-BC (Biomedical Translational Engineering Research Center of BUCT-CJFH) | XK2022-07 | Shi-Zhong Luo |
| Swedish Research Council | 2021-01094 | Jin-Ping Li |
| China Postdoctoral Research Foundation | Postdoctoral Fellowship Program GZC20230211 | Pengfei Pei |

The funders had no role in study design, data collection, and interpretation, or the decision to submit the work for publication.

### Author contributions

Xiaolin Yu, Guangfei Duan, Conceptualization, Methodology, Writing - original draft, Writing – review and editing; Pengfei Pei, Long Chen, Renji Gu, Wenrui Hu, Hongli Zhang, Yan-Dong Wang, Lili Gong, Lihong Liu, Methodology; Ting-Ting Chu, Supervision, Methodology, Writing – review and editing; Jin-Ping Li, Supervision, Writing - original draft, Writing – review and editing; Shi-Zhong Luo, Conceptualization, Supervision, Funding acquisition, Writing - original draft, Writing – review and editing

### Author ORCIDs

Shi-Zhong Luo (iD) http://orcid.org/0000-0002-4880-5962

## Ethics

All blood samples were obtained with informed consent, and the study was approved by the ethics review committee of China-Japan Friendship Hospital (2022-KY-050). All relevant ethical regulations of China-Japan Friendship Hospital and governmental regulations were followed.

The procedures for care and use of animals were approved by Beijing Municipal Science & Technology Commission, Administrative Commission of Zhongguancun Science Park (SYXK-2021-0056) and all applicable institutional and governmental regulations concerning the ethical use of animals were followed.

Joint public review: https://doi.org/10.7554/eLife.93871.4.sa1
Author response https://doi.org/10.7554/eLife.93871.4.sa2

## Additional files

### Supplementary files
• MDAR checklist

### Data availability

All data generated or analysed during this study are included in the manuscript and supporting files; source data files have been provided for Figures 1–6.

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
