## [Editor Report · eLife assessment]

How the triplicate interaction between chemokines with both GAGs and G protein-coupled receptors (GPCR) works and how gradients are created and potentially maintained in vivo are poorly understood. The authors provide **solid** evidence to show phase separation can drive chemotactic gradient formation. The paper is a **useful** advance in the field of chemokine biology.

---

## [Referee Report · Joint public review]

Chemokines are known to create chemotactic gradients and it is generally recognized that in order to create these gradients they need to bind to glycosaminoglycans (GAGs) on cells and in tissues. However, how the triplicate interaction between chemokines with both GAGs and G protein-coupled receptors (GPCR) works and how gradients are created and potentially maintained in vivo is poorly understood. In their manuscript, Yu et al investigated and showed in detail the ability of soluble and cell-bound GAGs to create gradients of the chemokine CCL5. They show in vitro in a modified leukocyte migration assay that soluble GAGs and GAGs on the tumor cell line THP-1 affect leukocyte migration. This useful work contributes to our in-depth understanding of the role of GAGs in chemokine gradient creation which is important for site-directed leukocyte and potentially tumor cell migration and as such is of potential interest for scientists studying immune responses in infection, inflammation, autoimmunity and tumor biology. In their reply to the comments of both reviewers they indicate that liquid-liquid phase separation (LLPS) was not detected at lower CCL5 concentrations. This is important information since, together with the tendency of CCL5 to form oligomers, it may indicate that oligomerization is crucial for LLPS. This info should at least be added to the discussion of the manuscript.

---

## [Author Response]

The following is the authors’ response to the previous reviews.

**Public Reviews:**

**Reviewer #2 (Public Review):**
Although the study by Xiaolin Yu et al is largely limited to in vitro data, the results of this study convincingly improve our current understanding of leukocyte migration.(1) The conclusions of the paper are mostly supported by the data and in the revised manuscript clarification is provided concerning the exact CCL5 forms (without or with a fluorescent label or His-tag) and amounts/concentrations that were used in the individual experiments. This is important since it is known that modification of CCL5 at the N-terminus affects the interactions of CCL5 with the GPCRs CCR1, CCR3 and CCR5 and random labeling using monosuccinimidyl esters (as done by the authors with Cy-3) is targeting lysines. The revised manuscript more clearly indicates for each individual experiment which form is used. However, a discussion on the potential effects of the modifications on CCL5 in the results and discussion sections is still missing.

Many thanks for the reviewer's suggestion. We fully agree it is important to clarify the potential issue of Cy3 labeling, and believe it is more suitable in the Materials and Methods section (line 312-314).

(2) In general, authors used high concentrations of CCL5 in their experiments. In their reply to the comments they indicate that at lower CCL5 concentrations no LLPS is detected. This is important information since it may indicate the need for chemokine oligomerization for LLPS. This info should be added to the manuscript and comparison with for instance the obligate monomer CCL7 and another chemokine such as CXCL4 that easily forms oligomers may clarify whether LLPS is controlled by oligomerization.

We are pleased by the help of the reviewers and accordingly inserted a brief discussion as suggested (line 240-246).

(3) Statistical analyses have been improved in the revised manuscript.

Thanks to the reviewer for his/her comment.